# Could Physical Therapy Interventions Be Adopted in the Management of Critically Ill Patients with COVID-19? A Scoping Review

**DOI:** 10.3390/ijerph18041627

**Published:** 2021-02-08

**Authors:** Carlos Bernal-Utrera, Ernesto Anarte-Lazo, Juan Jose Gonzalez-Gerez, Elena De-La-Barrera-Aranda, Manuel Saavedra-Hernandez, Cleofas Rodriguez-Blanco

**Affiliations:** 1Faculty of Nursing, Physiotherapy and Podiatry, University of Seville, 41009 Seville, Spain; cleofas@us.es; 2Fisiosur I+D Research Institute, 04630 Almería, Spain; clinicafisiosur@gmail.com (J.J.G.-G.); fisioelenacordoba@gmail.com (E.D.-L.-B.-A.); clinicasaavedra@yahoo.es (M.S.-H.); 3Doctoral Program in Health Sciences, University of Seville, 41009 Seville, Spain; anartelazo.ernesto@gmail.com; 4Centre of Precision Rehabilitation for Spinal Pain (CPR Spine), School of Sport, Exercise and Rehabilitation Sciences, University of Birmingham, Birmingham B15 2TT, UK; 5Department Nursing, Physiotherapy and Medicine, University of Almeria, 04120 Almería, Spain; 6Morphological and Socio-Health Sciences Department, University of Cordoba, 14071 Cordoba, Spain

**Keywords:** coronavirus, critical care rehabilitation, intensive care units, lung recruitment, physiotherapy, physical therapy modalities, severe acute respiratory syndrome

## Abstract

As part of COVID-19 consequences, it has been estimated that 5% of patients affected by this disease will require admission to the intensive care unit (ICU), and physical therapy techniques have been implemented in patients with other conditions admitted to ICU. The aim of the present study is to summarize all the available information about the implementation of physical therapy management in critically ill patients. From three clinical guidelines already published, we performed a search in PubMed, Scopus, ScienceDirect, and CINAHL, including systematic reviews, clinical guidelines, and randomized controlled trials, among others. Data extraction was performed independently by two reviewers. Quality assessment was developed through the AMSTAR-2 tool and PEDro Scale. A narrative synthesis was performed and 29 studies were included. The information extracted has been classified into four folders: ICU environment in COVID-19 (security aspects and management of the patient), respiratory physiotherapy (general indications and contraindications, spontaneously breathing and mechanically ventilated patient approaches), positional treatment, and exercise therapy (safety aspects and progression). The implementation of physiotherapy in patients affected with COVID-19 admitted to the ICU is a necessary strategy that prevents complications and contributes to the stabilization of patients in critical periods, facilitating their recovery.

## 1. Introduction

On 7 January 2020, the Chinese authorities identified a new virus, which was called SARS-CoV-2. It has been demonstrated that this virus can provoke various clinical symptoms, leading to a disease called COVID-19 [1]. At the date of 3 February 2021, according to the World Health Organization (WHO), there have been a total of 103,362,039 confirmed cases and 2,244,713 confirmed deaths attributed to COVID-19 [2]. Signs and symptoms of the disease can manifest as fever (98%), cough (76%), and myalgia or fatigue (44%) as the most typical; other symptoms described are sputum, headache, hemoptysis, vomiting, diarrhea, and dyspnea. In addition, many patients developed pneumonia, and one of the most typical complications was acute respiratory distress syndrome (ARDS) [3,4,5]. Due to its dramatic spread all over the world, many countries have been touched, leading to severe cases, deaths, and risk of healthcare and economic system collapse [6]. It is estimated that 80% of the patients will present mild symptoms (without hospital admission). The remaining 20% will need medical care, and 5% of them will require admission to the intensive care unit (ICU) [4,7]; consequently, the disease can result in morbidity, disability, and mortality.

Although no criteria have been specifically developed to determine when a patient with COVID-19 must be admitted to ICU, criteria from the American Thoracic Society/Infectious Disease Society of America (ATS/IDSA) have been adopted [8,9]. As part of ICU management, some techniques and approaches are well known, such as pharmacology, hemodynamic support, or invasive mechanical ventilation, among others [10]. While there are currently no studies reporting the effect of physical therapy treatment in critically ill patients with COVID-19, it has been implemented with success in some ICUs, and effectiveness in this kind of patients has been found in some systematic reviews [11,12]. The aim of physical therapy treatment in these patients is to avoid or reduce what is known as ICU-acquired weakness [13], which has been demonstrated to be a possible risk factor of increased morbidity and mortality [14]. Thus, due to the high number of patients admitted to ICU due to COVID-19, this pandemic has created a challenge for physical therapists to apply their knowledge and skills in these patients. Therefore, it is important to summarize all the available information related to the implementation of physical therapy management in critically ill patients, so that it could be included as part of the treatment of patients admitted to the ICU suffering from COVID-19.

The objectives of this study are intended to answer practical questions about current knowledge concerning clinical recommendations of physiotherapy worldwide, applied to those critically ill patients affected by COVID-19 or similar conditions, so that information can be adapted to this new disease, updating this knowledge until September 2020, and health professionals can access this clinical practical information in the most synthesized and updated way.

## 2. Materials and Methods

This review followed the guidance of the Preferred Reporting Items for Systematic Reviews and Meta-Analysis (PRISMA) Scoping Review Checklist [15] to report the study.

### 2.1. Eligibility Criteria

The PCC (Participant, Concept, Context) framework was used to develop eligibility criteria, according to the PRISMA Scoping Review Checklist [15].

#### 2.1.1. Participants

To be included in this review, papers needed to focus on critically ill patients with COVID-19 or similar conditions.

#### 2.1.2. Concept

This review was interested in different types of publications that assessed or proposed different ways of implementing physical therapy interventions in critically ill patients with COVID-19. It is important to point out that we did not focus on only one type of study design. Since COVID-19 is a recent disease, very little scientific literature has been published in this sense, so we also included those studies in which information provided explained how physical therapy has been implemented in critically ill patients with similar conditions, understood as those conditions where respiratory disorders could have led to an admission to ICU. In addition, we also admitted those articles where information provided could be used as guidelines of critically ill patients’ management, regardless of the underlying disease.

#### 2.1.3. Context

This review was not centered on a specific geographic location or on cultural factors but on the acute care of patients with COVID-19 who needed to be admitted to ICU. Our review was directed to synthesize all the information available to offer a broad overview of how physical therapy could be implemented in critically ill patients with COVID-19.

Therefore, from the three clinical guidelines already mentioned, which were our starting point, we analyzed the scientific literature and included those publications where the implementation of physiotherapy in the context of an ICU patient with COVID-19 or similar conditions was studied. We accepted different kinds of articles: systematic reviews, clinical guidelines, expert consensus, randomized controlled trials, prospective studies, and descriptive ones, among others; all of them providing information related to the scope of our review. Moreover, we looked for online information provided by official organizations; thus, we also included recommendations or guidelines offered by scientific societies, official technical committees, or similar bodies. On the other hand, exclusion criteria were (a) articles published before 2000, in order to collect updated information, (b) non-English/Spanish texts, (c) no available full-text article, (d) articles that analyzed physical therapy in an ICU setting but in very different conditions to COVID-19, such as the management of neonatal pathologies, continuous renal replacement management, Guillain–Barré syndrome, etc.

### 2.2. Information Sources

After a first scope review, we identified three published guidelines of Italian, Spanish, and Australian provenance [16,17,18]. Due to the shortage of publications related to the implementation of physical therapy in critically ill patients with COVID-19, we performed the first overview on these guidelines. Information extracted from them was our baseline, which allowed us to collect data about clinical practice rules in these patients. Furthermore, intending to amplify the contents of physical therapy interventions explained in these guidelines, we developed searches in four databases: PubMed, CINHAL, ScienceDirect, and Scopus. We also performed searches on websites of scientific societies, official technical committees, or similar bodies.

### 2.3. Literature Search

In the four databases mentioned before, we used multiple combinations of the following terms: “physiotherapy,” “physical therapy,” “intensive care units,” “rehabilitation,” “prone position,” “mobilization,” and “pulmonary rehabilitation” combined with the Booleans “OR” and “AND.”

The last search was developed on 12 September 2020. Two independent reviewers (C.B.-U. and E.A.-L.) developed this process. A first search was developed and, when concluded, both reviewers discussed results. If no agreement could be made, a third reviewer (C.R.-B.) was consulted, as a guaranteed method to avoid bias.

### 2.4. Charting Data

A data-charting form was developed by two reviewers (C.B.-U. and E.A.-L.) to determine which information to extract. The two reviewers independently charted the data, discussed the results, and continuously updated the data-charting form.

Data from eligible studies were charted using a standardized data-extraction tool designed for this study (Excel). The tool captured the relevant information on key study characteristics and detailed information on all categories of interest to organize data: authors, year of publication, country of origin, publication type or source, methodology, conceptual approach including terminology used, intervention or proposal included in each article, condition of interest, and quality of the evidence (if possible).

### 2.5. Data Items

We abstracted data on article characteristics, intervention studied (respiratory physiotherapy, mobilization/exercise therapy, or prone-positioning therapy), general recommendations, barriers and facilitators for the implementation of physical therapy on ICU context, recommendations and safety for these treatments, and characteristics for each type of patient (intubated, non-intubated, and extubated patients).

### 2.6. Critical Appraisal of Individual Sources of Evidence

This review compiles a wide variety of publications, including clinical guidelines, expert consensus, descriptive studies, etc. Therefore, performing quality assessment was not possible in every publication included, such as clinical guidelines and expert consensus. Nonetheless, we evaluated the quality of systematic reviews and randomized controlled trials through the AMSTAR-2 tool [19] and the PEDro scale [20], respectively. For descriptive studies, the STROBE [21] and MOOSE [22] recommendations were followed.

This procedure was performed by two reviewers (J.J.G.-G.) and (E.D.-L.-B.-A.) independently and later discussed. If any disagreement arose, a third reviewer (M.S.-H.) was consulted for taking a decision.

### 2.7. Synthesis of Results

A narrative synthesis of all the data extracted was performed. Since our goal is to offer an overview of information related to the physical therapy management of critically ill patients with COVID-19, we decided to divide this information into different sections, to clarify different issues concerning admission and management. Thus, first of all, we developed a section where we explained the inclusion criteria in the ICU and measures of prevention. Later, information about physical therapy management was synthesized and divided according to the different interventions studied: respiratory physiotherapy, exercise/mobilization therapy, and prone-positioning therapy.

## 3. Results

### 3.1. Study Selection

We identified a total of 3574 studies after reviewing databases, and after consulting the websites of societies, we found 17 guidelines/publications provided by them that could be of interest. After removing duplicates, 1806 publications remained for eligibility testing. Therefore, these sources of information were assessed against the criteria by two reviewers. Finally, 29 publications were selected and included in this review. A flow chart can be checked in Figure 1.

### 3.2. Relevant Data about Included Studies

A wide variety of publications were included as sources of information. Among them we can find five descriptive studies [4,23,24,25,26], one clinical protocol [9], ten clinical guidelines [16,17,18,27,28,29,30,31,32,33], one statement [34], one prospective study [35], three systematic reviews [36,37,38], four reviews [39,40,41,42], two expert consensuses [43,44], four randomized controlled trials [45,46,47,48], and one comparative study [49]. Information can be found in Table 1.

### 3.3. ICU Environment in COVID-19

Although no criteria have been established in COVID-19 patients, Sequential Organ Failure Assessment (SOFA) and ATS/IDSA criteria can be used as a guide to establish which patients must be admitted to the ICU [8,9]. The subjects must meet at least three minor criteria or one major criterion of ATS/IDSA [34]. Special attention must be given to those subjects considered high risk: pregnant patients, people older than 60 years, and those patients with other previous pathologic processes such as lung disease, chronic kidney disease, arterial hypertension (AHT), cardiovascular disease, diabetes, immunocompromised, etc. [18,23].

Physical therapy management depends on the stabilization of vital values and the decision of the intensivist team. The main values of reference can be checked below [9,39]:Breathing frequency <24 respirations per minute (RPM)Resting heart rate <50% age-predicted maximal heart rateBody temperature <38 °CBlood pressure <20% variability recentlyPaO_2_/FIO_2_ >300, SpO_2_ >90%, and < 4% recent decrease in SpO_2_ (if not previous respiratory failure)A stable level of consciousnessElectrocardiogram (ECG) normal (i.e., no evidence of myocardial infarction or arrhythmia)

### 3.4. Important Items before Applying Physical Therapy Interventions

All physical therapists must be aware of the importance of considering the risk/benefit of each of their decisions. Many techniques implemented by physiotherapists in ICU are considered of high risk because of aerosol and microdroplet generation, such as aerosol therapy, use of humidifiers, pep bottles, or sputum inductions [18]. Exposition to COVID-19 patients must be reduced to the minimum, based on two reasons: preventing infection and avoiding an irrational waste of important materials, such as personal protective equipment (PPE) [16,17,18].

All procedures detailed in this article, especially those related to breathing physiotherapy, are required to be performed with trained, experienced professionals. It is of vital importance to establish a good diagnosis and to maintain good communication between different professionals from a multidisciplinary team since not all procedures and techniques are suitable for all patients. The implementation of these techniques in an unspecific and generalized way could lead to more serious problems in critically ill patients.

Three procedures have been detailed as treatment options in ICU patients: respiratory physiotherapy, prone positioning, and exercise therapy. A guide can be checked in Figure 2.

### 3.5. Respiratory Physiotherapy

In COVID-19 patients, some situations can be present. In patients with dry, nonproductive cough, fever, and/or without changes in thorax radiography, breathing techniques are not indicated [18], in the same way as in patients with pneumonia, nonproductive cough, and SpO_2_ <88% [40].

Generally, physical therapy treatment is indicated in the following circumstances: (a) mild symptoms and/or pneumonia, and co-existing respiratory or neuromuscular comorbidity, and difficulties with secretion clearance or (b) mild symptoms and/or pneumonia, with difficulty to clear or inability to clear secretions independently. Finally, in the case of (c) severe symptoms suggesting pneumonia or infection of the lower respiratory tract, physical therapy could be considered for airway clearance, especially if weak cough, productive, and/or evidence of pneumonia on imaging and/or secretion retention [18]. The procedures recommended are detailed below. All these procedures, if possible, should be applied in isolated rooms, with negative pressure and PPE [18].

#### 3.5.1. Non-Intubated and Extubated Patients

The following techniques have been recently proposed for the expulsion of secretions, to increase ventilatory capacity and to improve the stimulation of the respiratory muscular system [18,27].

##### Active Cycle of Breathing Technique (ACBT)

The method of ACBT consists of, at first, deep breathing to collect mucus from peripheral airways at the bottom of the lungs toward the upper airways, and then performing a huff cough to expel secretions [27].

Positioning/gravity-assisted drainage combined with manual techniques (expiratory vibrations and percussions)

With changes in posture, the objective is the clearance of bronchial secretions and it consists of using the effect of gravity over the secretions from the periphery of the lung toward the central upper airway. In addition, the ventilation of different lung areas is intimately related to posture, and this effect is used to avoid the blocking of bedridden subjects. Postural management is complemented with manual techniques such as expiratory vibration and percussion, which are performed only during the expiratory period [27].

##### Mechanical Insufflation–Exsufflation

This technique aims to clean secretions by the application of a gradual positive pressure to the airway, then rapidly changing toward negative pressure. The fast change in pressure produces high expiratory flow, which mimics a natural cough. The mechanical insufflator–exsufflator provokes a deep insufflation (positive pressure of 30 to 50 cm H_2_O) followed, immediately, by a deep exsufflation (negative pressure of −30 to −50 cm H_2_O) [27].

##### Directed Cough or Manually Assisted Cough

The teaching of voluntary coughing is known as “directed cough”. It can be performed at a high volume (initiated by total pulmonary capacity) or at a low volume (initiated by the residual functional capacity). Manually assisted cough helps to clear secretions by applying a positive pressure to fill the lungs, then quickly switching to a negative pressure to produce a high expiratory flow rate, simulating a cough. It is indicated in patients with high levels of sedation or difficulty in voluntarily coughing [27].

##### Huff Coughing

This technique uses the gas compression and is performed through one or two forced expirations. It starts with a medium lung volume and is continued until the residual volume is achieved without puckering lips, with the aim of producing a cough and propelling secretions. The advantage is that it provokes less fatigue, it has less tendency to develop bronchospasm, and produces less dynamic compression of airways because transpulmonary pressure is reduced [27].

##### Manual Hyperinflation (MHI)

It allows the manual control of insufflated volumes of air and the live perception of possible intra-lung resistances [27].

##### Positive Expiratory Pressure Devices

Positive end-expiratory pressure (PEEP) or oscillating positive expiratory pressure (OPEP) are physical therapy techniques implemented in the thoracic region. When the therapy is performed through PEEP, expiration provokes a positive pressure in breathing airways, which promotes the opening of small and deep airways. On the other hand, OPEP therapy consists of the combination of PEEP with high-frequency oscillations. It involves breathing with a slightly active expiration against an expiratory resistance through a device [27].

##### Inspiratory Musculature Training (IMT)

It has been demonstrated that invasive ventilation produces a reduction in muscular respiratory function [35]. Because of that, it is important to exercise or to apply techniques that could lead to an improvement of inspiratory volumes in those patients who needed invasive ventilation [28]. Training of inspiratory musculature through threshold pressure training generates significant benefits in patients who have been treated with mechanical ventilation [36]. Intensity and dosage must be adapted to patients, and no evidence exists about parameters that could lead to better results [37]. This procedure is contraindicated in the acute phase, so its implementation is recommended post-extubation [16,17,18].

#### 3.5.2. Intubated Patients

##### Ventilator Hyperinflation (VHI)

As an alternative to MHI, VHI has been proposed. Its purpose is to deliver higher tidal volume breaths than normal to enhance secretion removal, improve oxygenation, and ameliorate lung compliance. Hemodynamic changes associated with decreased cardiac output and increased intracranial pressure have been recognized as possible complications [16,18,27].

##### Humidification

The analyzed clinical guidelines do not recommend its use in patients with COVID-19 [15,16,17].

##### Aspiration

Prior to its implementation, other techniques of respiratory physiotherapy should have been applied to mobilize secretions. Aspiration is developed through closed systems to secretions, following strict security measures. The secretions’ aspiration is developed through the introduction of a probe to the bronchial tree, pharynx, or mouth, to aspire the secretions that obstruct it, preventing the eventual formation of mucous plugs [16,17,18].

These procedures are implemented to maintain the airways’ permeability, to clean airways, to prevent the formation of mucous plugs, to avoid the blocking of the intubation probe, to expel secretions, to clean the bronchio-pulmonary region, and to avoid the risk of infectious processes in patients at risk [27].

### 3.6. Positional Treatment

#### 3.6.1. Non-Intubated and Extubated Patients

Some positions have been studied to clarify their effect on ICU patients: semi-sitting (40°–60°) or sitting, in addition to decubitus lateral, semi-prone, and prone positioning [16]. Postural changes can both positively and negatively modify the ventilation/perfusion (V/P) index. An individualized control must be performed in each patient. As a premise, efforts performed by the patient to maintain different postures should be minimized as much as possible. In addition, slump positions with thoracic blocking must be avoided [16].

#### 3.6.2. Intubated Patients

Prone positioning is highly recommended in critically ill patients with invasive ventilation infected by COVID-19. Three guidelines consulted recommend this position for 12–16 h per day [16,17,18]. A systematic review demonstrated that prone positioning can improve the oxygen level of arterial blood in patients with ARDS admitted to the ICU [31]. Oxygenation is produced by the optimization of lung recruitment, improving the V/P index [41]. However, this maneuver also entails some risks and can imply transitory changes in oxygen saturation and hypotension. Thus, exhaustive control is required [16,41,43]. In case of a decrease in the V/P index higher than 20% when compared to supine positioning, or other serious complications, an urgent intervention must be carried out [16].

The maneuver is not indicated in all patients. Two main contraindications have been established: spinal instability and/or increase in intracranial pressure (not frequent in COVID-19 patients). Other relative contraindications are pregnant patients, hemodynamic instability, simple fractures, or open abdominal wounds [41,43]. In addition, positional change must be only performed when enough technical resources are available, and by professionals with experience [18]. The Intensive Care Society and Faculty of Intensive Care Medicine published a guideline where the detailed procedure can be found, in addition to requirements for a correct application of the positioning maneuver [29]. This procedure must be performed with caution since several risks during this procedure have been explained, such as accidental extubation or catheter displacement, shoulder luxation, brachial plexus injuries, or pressure ulcers [29,30,41]. The clinical guidelines consulted do not consider alternatives for those patients in whom prone positioning is contraindicated [16,17,18,29].

### 3.7. Mobilization and Exercise Therapy

In patients who could develop a significant functional deficit and/or intensive care unit–acquired weakness (ICUAW), mobilization and exercise therapy are indicated. These interventions range from passive and active mobilization to rehabilitation of physiological movements (sit at the bed’s side, sit-to-stand, steps, etc.) [18].

General considerations before applying mobilization and/or exercise therapy

First of all, based on prevention rules, it is important to plan the identification of the minimum necessary personnel to develop the activity and to make sure that the material that will be used is cleaned and disinfected [18].

Early mobilization is highly recommended, but security must be considered as a priority [18]. In fact, patient safety is one of the most commonly reported barriers to delivering early mobilization [31]. Mobilization and therapeutic exercise must be considered according to the physiological variables of patients, discussed by a multidisciplinary team, and constantly monitored during its implementation [18,32].

In- and out-of-bed exercises are not contraindicated in patients with endotracheal or tracheostomy tubes [44]. However, some respiratory parameters must be monitored: (a) with a fraction of inspired oxygen higher than 0.6, precautions for both types of exercises must be established, and mobilizations must be done cautiously; (b) in case of a percutaneous oxygen saturation smaller than 90%, out-of-bed exercises are contraindicated, while in-bed mobilizations, if performed, must be done with caution and gradually [44]. It is of vital importance to establish a progression not only about which kind of exercises/mobilization could be implemented but also about the patient’s position during these procedures [24], since it has been documented that different positions change the V/P index, so it would modify the capacity of patients to perform exercises, and also their security [18,24]. When a patient’s physiological response exceeds these parameters, mobilization should be stopped and restarted after a period of rest and monitoring. If an adverse event happened, mobilization must be ceased immediately, and the patient should be referred for an urgent medical review; a daily assessment must be developed to determine which mobilization technique must be performed [32].

#### 3.7.1. Non-Intubated Patients

As explained before, a deep assessment has been proposed prior to physical therapy mobilizations. At first, the therapist must evaluate the level of alertness and the ability to follow instructions. Later, an evaluation of the presence of any barriers to mobilization must be developed; this includes the possibility that the organ support equipment could make mobilization difficult. If these first steps are not overcome, passive mobilization could be performed [32], with the aim of reducing skin lesions and immobilization sequelae [16]; in addition, it has been suggested that passive mobilization of lower limbs is well tolerated and could produce a short-term effect in the reduction of inflammatory cytokine levels [25], which could be very interesting in patients with COVID-19. Moreover, even with a sedated patient, therapists could try to assist the patient to stay in a sitting position, to achieve clinical stability and upright sitting, minimizing orthostatic intolerance [24]. If the patient is awake but barriers to mobilization are present, physiotherapists could start with an in-bed exercise program aimed to maintain strength and range of motion [32], both in the upper and lower limbs, e.g., cycling in bed [45].

When the patient shows no problem at these first steps, a physical assessment must be performed prior to starting active mobilization. If the patient does not have safe unsupported sitting and <3/5 strength (Oxford scale) in lower limbs [32], in addition to being hemodynamically stable [24], he or she could perform some in-bed activities such as sitting balance, tilt table, strengthening, etc. On the other hand, when the patient achieves positive results in this assessment, we have two options. If the subject cannot stand with the assistance of two professionals, supported weight-bearing should be trained (gait harness, sit-to-stand practice, etc.). If that is possible, physical therapists would proceed with active weight-bearing, progressing daily to achieve functional goals [32].

#### 3.7.2. Intubated Patients

In these patients, it has been argued that intubation is not in itself a contraindication to early mobilization [31], and it has been demonstrated to be safe and feasible [38,49]. Nonetheless, physiotherapists must be aware of maintaining airway security; if necessary, another professional could help the physical therapists to prevent disconnections of the airway [18]. Indeed, it has been argued that similar activities, such as mobilization out of bed, can be performed both in non-intubated and intubated patients [42], always paying attention to physiological variables and limiting factors.

To make reference to the mobilization procedure, a mnemonic rule “Plan B” was developed [24], consisting of: preparation, leader, airway and emergency equipment, number of staff, and backup planning. In summary, there are some key points to consider prior to and during mobilization: to prepare and plan the mobilization with the rest of the multidisciplinary team (timing, equipment required, and physical environment are important to be checked); to identify a leader and the specific roles; to check the function, clearance, and security of artificial airways; to establish the professionals needed for mobilization; to prepare a backup plan, which may be implemented if any adverse event happens [31,32,44].

Since many factors could influence this procedure, it has been argued that the clinical decision procedure could be based on ranking the ability of patients to cope with all these factors [26]. It has been proposed that, if the patient can perform sit-to-stand and static standing at the bedside and vital signs, level of alertness, and trunk control are controlled, a physical therapist could proceed with activities in standing position, transferring to chair and gait training [33].

In relation to the frequency of exercise sessions, a study evaluated 15 min/day in mechanically ventilated patients, achieving positive results when a 6-min walking test (6MWT) was assessed [46], while in another study a frequency of five times per week was implemented also in mechanically ventilated subjects, without specifying time per session [47].

#### 3.7.3. Extubated Patients

Patients who have weaned from mechanical ventilation and can breathe independently should continue improving respiratory capacities [29], as well as other physiological functions such as neuromuscular or cardiovascular activities. Less has been investigated in this kind of patient, although continuing with the aim of achieving functional goals seems logical.

It has been studied that similar exercises to those performed in mechanically ventilated patients, such as arm and leg active movements, sit-to-standing, or marching in place, and with a frequency of 2 × 15 min/day in extubated patients achieved positive results when 6MWT was assessed [48]. On the other hand, adding a supported arm training exercise (20 min/day in arm ergometer) to a general physiotherapy program (45 min/day every 3 days) may be useful and safe in the recovery of recently weaned critically ill patients [49].

## 4. Discussion

The COVID-19 global pandemic has led to a high increase in patients admitted to ICUs. It is important to point out that physical therapists’ competences are different according to each country’s regulation, so functions will differ. In addition, health policies can also vary; thus, some of the procedures described in this review could be not recommended in some countries. Moreover, in-hospital requirements and saturation could make the implementation of these procedures difficult. During this pandemic, we have observed that acute care and ICUs have become overwhelmed in many countries. In that sense, according to a recent qualitative study performed in Spain, physical therapists have been obligated to help and develop other procedures not directly related to their profession [50], which indirectly hinders the application of these therapies [51]. Therefore, clinicians should consider deeply whether these procedures are indicated in the context of their clinical setting or could be delayed. Nonetheless, despite these objections, physical therapy treatment has been demonstrated to be effective as part of ICU management [11,12,52].

Thus, the aim of our review was to analyze and collect information available in the current scientific literature about physical therapy management in critically ill patients with COVID-19 or with similar respiratory conditions. The implementation of physical therapy in non-invasive mechanically ventilated patients was not considered. This is due to the high failure rate, the risk of aerosol generation, and the requirement of invasive mechanical ventilation in these patients [18]. The Italian guideline suggests a single attempt at non-invasive mechanical ventilation, with a duration of one hour [16], but this could be due to the shortage of material needed for this kind of interventions. Future publications should clarify this point.

### 4.1. Respiratory Physiotherapy

An interesting fact is that the Italian guideline [16] does not recommend the implementation of breathing physical therapy. In this guideline, some maneuvers such as lung-recruitment maneuvers and airway-clearance techniques are mentioned as possible techniques to implement when considered strictly needed, but to their knowledge, they argue that they are, respectively, dangerous and not frequently required in COVID-19 patients [16]. Nonetheless, breathing physical therapy is recommended in the two other guidelines [17,18]. We think that it could be explained by the risk/benefit in relation to virus infection. Possibly, in the Italian guideline, the risk of contagion considered is higher than the benefits produced by breathing physical therapy, and that is the reason why they argued that it should not be implemented. However, in the two other guidelines, they argued that this decision should be taken by the multidisciplinary team because benefits can be significant [17,18]. In fact, the Australian guideline refers us to comply with international, national state, and/or hospital guidelines for infection control, following the example of WHO guidelines [53]; nonetheless, it is necessary to consider that resources such as personal protective equipment and requirements are not inexhaustible [54], so the decision to implement physical therapy must be deeply considered, as it has been argued that the COVID-19 outbreak will cause many limitations in the rehabilitation department and other hospital facilities [55,56]. On the other hand, the implementation of inspiratory musculature training (IMT) in extubated patients can court controversy. The deterioration of ventilation parameters after mechanical ventilation has been proved [28]; however, none of the consulted guidelines proposes the implementation of IMT after mechanical ventilation, independently of the stage of rehabilitation [16,17,18]. The implementation of IMT in ICU patients post-extubation could be considered a little risky due to the delicate condition of the patient and the possibility of inducing changes in vital values. Nonetheless, the benefits of the activation of inspiratory muscles are significant [28,29,30]. The IMT must be implemented in all patients under security criteria and the approval of the multidisciplinary team. This will modify the moment of therapy implementation; nevertheless, although it can be postponed in some patients at risk, at least it should be implemented in all patients in the subacute stage because of its impact on health and the future quality of life of patients [28,29,30]. Another key about breathing physiotherapy is assisted postural drainage with manual techniques (percussion and vibrations). In fact, it is mentioned and indicated in the Australian guideline [18]. However, nowadays there is controversy about its efficacy; there are very few and outdated studies, and a Cochrane review from 2016 in pediatric patients discourages its use due to its inefficiency and because of the associated side effects [57]. These techniques are obsolete in current clinical practice.

### 4.2. Positional Treatment

In relation to the patient’s position, while the patient does not require mechanical ventilation, it is recommended to alternate positions generating a dynamic positioning. It is important to reduce at minimum the efforts to maintain the posture and to monitor oxygen saturation and perceived dyspnea [16,17,18]. We found that the most controversial issue is observed in intubated patients. Prone positioning for 12–16 h per day is recommended in all guidelines consulted and supported by several publications [16,17,18,31,32,33,34,35]. How to perform this maneuver and how to avoid complications due to pressure or immobility in a prone position have also been published [16,35]. However, large periods of time in a prone position could lead to difficulty to implement mobilization therapy. Interdisciplinary team cooperation and training are required to prevent complications related to rest. In patients in whom prone positioning is contraindicated, although nothing has been published about this procedure, it could be possible to consider the implementation of other positions, such as semi-prone or lateral decubitus. Medical teams could consider these options after following clinical reasoning based on security. Since changes in position can both positively and negatively modify the V/P index, individualized control of the patient must be carried out.

### 4.3. Mobilization and Exercise Therapy

The three guidelines make reference to the necessity of implementing mobilization therapy in ICU patients. The Italian guideline makes a light reference to passive mobilization to reduce skin lesions and immobilization sequelae, and to the necessity of discussing with the multidisciplinary team to start an early active-mobilization program [16]; the Spanish Society of Pulmonology and Thoracic Surgery (SEPAR) guideline encourages to reduce possible complications in ICU patients by early passive and active mobilization [17]; finally, the Australian guideline proposed that any patient at significant risk of developing or with evidence of significant functional limitations could be referred to physiotherapy, taking precautions to avoid contagion [18]. It is also necessary to take into account that there are some barriers to early mobilization in these subjects: patient related, structural, cultural, and process related. Thus, it is important to identify practical strategies to overcome these barriers [58].

On the one hand, it has been shown that just passive mobilization can produce positive changes in the immune system [40]; on the other hand, it has been observed that active mobilization/exercise therapy can produce changes both in the short and long term [59,60]. However, we find that mobilization and exercise therapy have received less consideration than in other studies [38,39,40,41,42,43,44,45,46,47,48,49], although reported adverse events are minimum and potential benefits are high [36]. This might be due to several reasons: the saturation of ICUs, the absence of evidence of these procedures in COVID-19 patients, and/or because the risk of virus contagious is very high. Nonetheless, when protective personal equipment and other security requirements are available, we find that it would be very interesting to give more importance to these procedures, given the effectiveness that has been demonstrated [50], which could offer the possibility to reduce the length of stay in ICU survivors [61]. In addition, physical therapy management should be continued after patients are discharged from ICU to reduce ICU-acquired weakness. For this purpose, management of patients with post-intensive care symptoms should continue rehabilitation in the hospital, but intensive care follow-up programs in post-ICU clinics or at home should also be considered [62,63], as has already been done [64]; in that sense, telerehabilitation appears as a feasible option [65,66]. That is why we developed a summary of mobilization progression based on different proposals [37,39,41,46] that can be checked in Figure 3. Nevertheless, we consider that it is very important to follow the guidelines’ recommendations about security.

This review aimed to show all the information available in relation to physical therapy management in ICU patients with COVID-19, with the intention of proportioning the most possible comprehensive review, detailing different aspects of this management. In that sense, we think this review could be used by clinicians around the world to improve their healthcare provision to critically ill patients, which could lead to reducing the length of stay and fewer complications related to intensive care.

### 4.4. Study Limitations

We present a theoretical guideline that is based on current scientific literature. First of all, we find important to point out that we included in our review different kinds of articles, such as clinical guidelines, that could not be assessed in terms of quality. We are conscious that it could become a limitation; nonetheless, our aim is not to assess the effectiveness, or ineffectiveness, of physical therapy procedures in critically ill patients admitted to ICU with COVID-19, and we found that clinical guidelines could offer us information not yet available in scientific sources and of interest for clinicians. Secondly, procedural details can differ from those carried out in hospitals because of system requirements and saturation. Due to the fact that not all patients are going to get benefits from these procedures, monitoring and joint decision-making by multidisciplinary teams and individualized care are required. Finally, although COVID-19 shares similarities with other respiratory conditions, we still do not know the nature of this disease, and different presentations can be found; therefore, these procedures should be implemented with caution and following a clinical reasoning process.

## 5. Conclusions

The implementation of physiotherapy in patients affected with COVID-19 admitted to the ICU is a necessary strategy, since it prevents complications and contributes to the stabilization of patients in critical periods, facilitating their recovery. Our review found that treatment in the ICU could be based on three main treatment modalities. At first, respiratory physiotherapy has been found to favor pulmonary ventilation, the mobilization and excretion of secretions, and the stimulation of respiratory muscles. Secondly, positional treatment has been proposed to increase the V/P index. Finally, exercise therapy could improve immune function and reduce complications, favoring functional recuperation in patients affected by COVID-19 admitted to ICU. A clinical reasoning process must be followed before implementing these procedures, and physical therapy treatment should be adapted to the context of the clinical setting.

## Figures and Tables

**Figure 1 ijerph-18-01627-f001:**
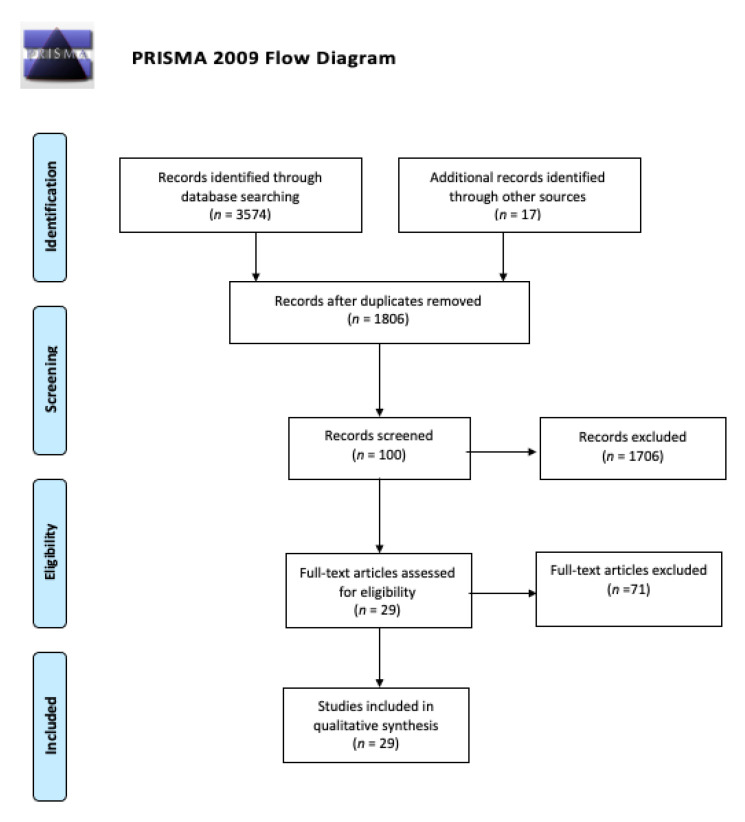
Preferred Reporting Items for Systematic Reviews and Meta-Analysis (PRISMA) flow diagram.

**Figure 2 ijerph-18-01627-f002:**
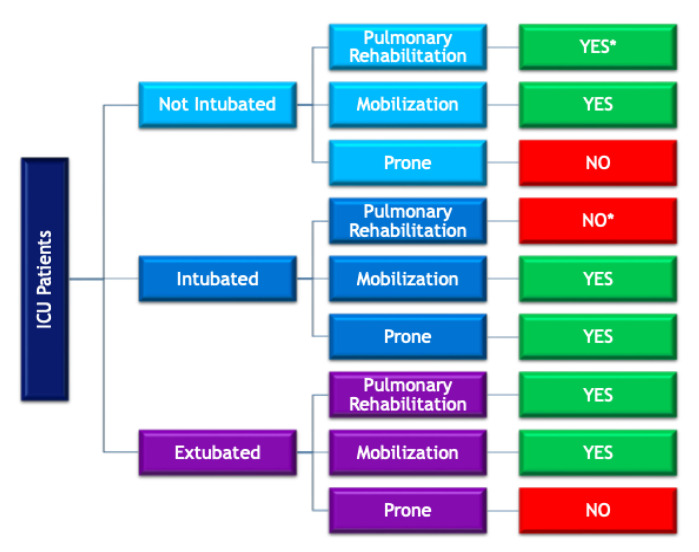
Main topics for the implementation of physical therapy in intensive care unit (ICU) patients with COVID-19.

**Figure 3 ijerph-18-01627-f003:**
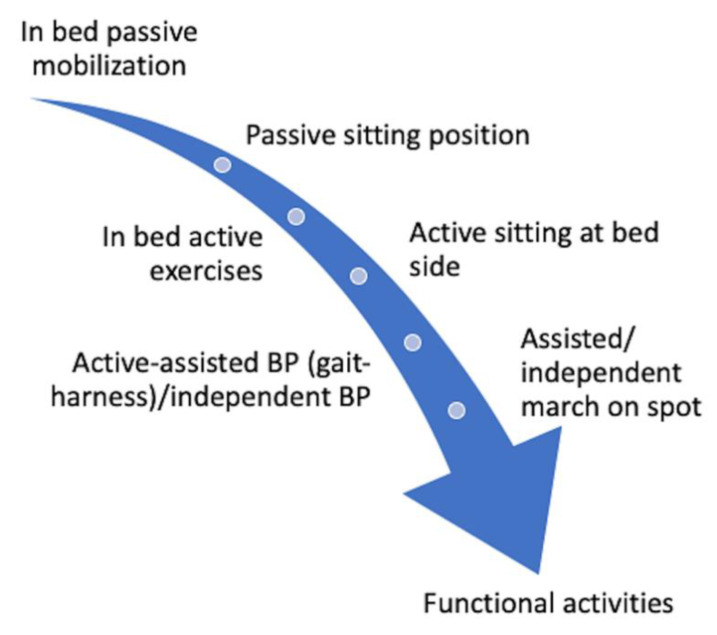
Active exercise progression model.

**Table 1 ijerph-18-01627-t001:** Studies characteristics and quality assessment.

Author′s Name/Society	Year of Publication	Journal Name	Design	Condition of Interest	Intervention/Assessment ^1^	Quality ^2^
Li et al. [23]	2020	The New England Journal of Medicine	Descriptive study	COVID-19 ^3^	-	-
Guan et al. [4]	2020	medRxiv preprint	Descriptive study	COVID-19	-	-
Spanish Government [9]	2020	-	Clinical Protocol	COVID-19	-	-
Lazzeri et al. [16]	2020	Monaldi Archives for Chest Disease	Clinical Guideline	COVID-19	-	-
SEPAR [17]	2020	-	Clinical Guideline	COVID-19	-	-
Thomas et al. [18]	2020	Journal of Physiotherapy	Clinical Guideline	COVID-19	-	-
Griffith et al. [34]	2007	American Journal of Respiratory and Critical Care Medicine	Statement	Nontuberculous mycobacterial disease	-	-
College of Physical Therapy from Andalucia (Spain) [27]	2020	-	Clinical Guideline	Respiratory Interventions	-	-
Medrinal et al. [35]	2016	Critical Care	Observational Cohort Study	Mechanically ventilated critically ill patients	Respiratory weakness	-
Bisset et al. [28]	2019	Australian Critical Care	Clinical Guideline	Mechanically ventilated critically ill patients	Inspiratory muscle training	-
Elkins et al. [36]	2015	Journal of Physiotherapy	Systematic review	Mechanically ventilated critically ill patients	Inspiratory muscle training	9/13
Munshi et al. [37]	2017	Annals of American Thoracic Society	Systematic review	ARDS ^4^	Prone positioning	13/16
Gattinoni et al. [41]	2013	American Journal of Respiratory and Critical Care Medicine	Narrative Review	ARDS	Prone positioning	-
Messerole et al. [43]	2002	American Journal of Respiratory and Critical Care Medicine	Expert Commentary	ARDS	Prone positioning	-
Intensive Care Society [29]	2019	-	Clinical Guideline	Intensive care patients	Prone positioning	-
Hodgson et al. [31]	2018	Critical Care	Clinical Guideline	Intensive care patients	Mobilization	-
Green et al. [32]	2016	Journal of Multidisciplinary Healthcare	Clinical Guideline	Intensive care patients	Mobilization	-
Hodgson et al. [44]	2014	Critical Care	Expert Consensus	Mechanically ventilated critically ill patients	Mobilization	-
Bassett et al. [24]	2012	Intensive and Critical Care Nursing	Descriptive study	Intensive care patients	Mobilization	-
Amidei et al. [25]	2013	American Journal of Critical Care	Descriptive study	Mechanically ventilated critically ill patients	Mobilization	-
Burtin et al. [45]	2009	Critical Care Medicine	RCT ^5^	Intensive care patients	Early exercise	4/10
Engel et al. [49]	2013	Critical Care Medicine	Comparative study	Intensive care patients	Mobilization	-
Nydahl et al. [38]	2017	Annals of American Thoracic Society	Systematic review	Intensive care patients	Mobilization	14/16
Ntoumenopoulos et al. [42]	2015	Intensive Critical Care Nursing	Narrative Review	Mechanically ventilated critically ill patients	Mobilization	-
Berney et al. [26]	2019	Archives of Physical Medicine and Rehabilitation	Descriptive study	Intensive care patients	Out-of-Bed exercise	-
Engel et al. [33]	2013	Physical Therapy	Clinical Guideline	Intensive care patients	Mobilization	-
Denehy et al. [46]	2013	Critical Care	RCT	Intensive care patients	Exercise	7/10
Chiang et al. [47]	2006	Physical Therapy	RCT	Mechanically ventilated critically ill patients	Exercise	6/10
Porta et al. [48]	2005	Chest	RCT	Recently weaned patients	Exercise	5/10

^1^ Due to study design, no intervention was performed in some studies; ^2^ Some studies could not be assessed in terms of quality due to study design; ^3^ COVID-19: Coronavirus Disease 2019; ^4^ ARDS: acute respiratory distress syndrome; ^5^ RCT: randomized controlled trial.

## Data Availability

Not applicable.

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
