# Peer review of "Could Physical Therapy Interventions Be Adopted in the Management of Critically Ill Patients with COVID-19? A Scoping Review"

_ijerph, 2021, doi:10.3390/ijerph18041627_

Round 1

Reviewer 1 Report

I wish to thank the authors for submitting their manuscript which discusses the possibility of utilising physical therapy for ICU patients with COVID. The authors draw from COVID-19 guidelines and other sources which include respiratory conditions similar to COVID and/or critical patients with ICU. They offer several potential avenues for physical therapy in these conditions and provide rationale for these. Naturally COVID-19 is a very important issue in all our lives currently, and the authors have addressed an issue which would help improve the quality of service and help aid the recovery of individual patients. 

Whilst I feel that the authors have added caveats to their findings, I do wonder how feasible this would be in the current pandemic scenario. Specifically, within my country ICUs and hospitals are struggling significantly (as are most countries) to cope with the current number of cases and admissions. I would therefore suggest that whilst physical therapy is key for some patients, I feel the additional stress both on staff and equipment requirements would prevent this from being practical in the given environment. I would therefore suggest that these approaches are more applicable moving forward when the pandemic has lessened and COVID-19 incidence becomes similar to that of seasonal flu, for example. I would therefore suggest the addition of such a sentence of section within the concluding sentences. 

I would also welcome discussion surrounding the idea of physical therapy and rehabilitation following discharge from the ICU - in particular, that of exercise intervention to counteract the ICU acquired weakness. 

One final comment, is that I feel we do need to be somewhat cautious with COVID-19, inherently due to the infectious nature of the virus, but also because of lack of understanding for this very new disease. Whilst it may share similarities to other conditions which require mechanical ventilation and/or ICU admission, we currently cannot fully appreciate the systemic effect of COVID. 

Aside from the above, overall the paper reads okay, but there are a few typing errors which need to be checked. In addition, I would suggest asking someone external read over the paper as there are areas throughout which are a little 'wordy' and require attention. Whilst I understand what is being said, small alterations would make it easier to read.

I have a few small comments to add below: 

Line 48: remove '.' in middle of the sentence 

Line 100: Presumably organisms is a typing error 

Lines 121 and 134: Please provide details as to which databases and which tools have been utilised. 

Reviewer 2 Report

This is an interesting review which deserves publication. In general, the flow of the paper should be improved, mostly introduction section. Some parts are slightly difficult to follow. For instance, those related to symptoms in the introduction. 

Please update data of COVID-19 infected people to 2021, not September 2020. 

The eligibility criteria, as this is a sopping review, could be written following the PCC mnemonic, which is the same one than PICO for SR. This would help reading this section. 

which data base were searched for? 

There are several abbreviations which are not properly described the first time of use. Please review this. 

Authors could include a figure with the main topics found in the review for a better visualization of all domains. It is highly difficult to follow the text with a figure. 

Please, numbering all subheadings in the discussion section not just the limitation one. 

Please include figure appendix into there main text. 
